# Proofing Direct-Seeded Rice with Better Root Plasticity and Architecture

**DOI:** 10.3390/ijms22116058

**Published:** 2021-06-04

**Authors:** Siddharth Panda, Prasanta Kumar Majhi, Annamalai Anandan, Anumalla Mahender, Sumanth Veludandi, Debendranath Bastia, Suresh Babu Guttala, Shravan Kumar Singh, Sanjoy Saha, Jauhar Ali

**Affiliations:** 1Crop Improvement Division, Indian Council of Agricultural Research (ICAR)-National Rice Research Institute (NRRI), Cuttack 753006, Odisha, India; siddhu0410@gmail.com (S.P.); sumanth479@gmail.com (S.V.); 2Department of Plant Breeding and Genetics, Odisha University of Agriculture & Technology, Bhubaneswar 751003, Odisha, India; debendranath.bastia@gmail.com; 3Department of Genetics and Plant Breeding, Institute of Agricultural Sciences, Banaras Hindu University (B.H.U.), Varanasi 221005, Uttar Pradesh, India; prasantakumarmajhi53@gmail.com (P.K.M.); shravanbhu1964@gmail.com (S.K.S.); 4Rice Breeding Platform, International Rice Research Institute (IRRI), Los Baños, Laguna 4031, Philippines; m.anumalla@irri.org; 5Department of Genetics and Plant Breeding, Naini Agricultural Institute, Sam Higginbottom University of Agriculture, Technology and Sciences (SHUATS), Prayagraj 211007, Uttar Pradesh, India; sureshgpb03@gmail.com; 6Crop Production Division, Indian Council of Agricultural Research (ICAR)-National Rice Research Institute (NRRI), Cuttack 753006, Odisha, India; ssahacrri@gmail.com

**Keywords:** direct-seeded rice, root system architecture, root plasticity, quantitative trait loci, genes

## Abstract

The underground reserve (root) has been an uncharted research territory with its untapped genetic variation yet to be exploited. Identifying ideal traits and breeding new rice varieties with efficient root system architecture (RSA) has great potential to increase resource-use efficiency and grain yield, especially under direct-seeded rice, by adapting to aerobic soil conditions. In this review, we tried to mine the available research information on the direct-seeded rice (DSR) root system to highlight the requirements of different root traits such as root architecture, length, number, density, thickness, diameter, and angle that play a pivotal role in determining the uptake of nutrients and moisture at different stages of plant growth. RSA also faces several stresses, due to excess or deficiency of moisture and nutrients, low or high temperature, or saline conditions. To counteract these hindrances, adaptation in response to stress becomes essential. Candidate genes such as early root growth enhancer *PSTOL1*, surface rooting QTL *qSOR1*, deep rooting gene *DRO1*, and numerous transporters for their respective nutrients and stress-responsive factors have been identified and validated under different circumstances. Identifying the desired QTLs and transporters underlying these traits and then designing an ideal root architecture can help in developing a suitable DSR cultivar and aid in further advancement in this direction.

## 1. Introduction

Rice (*Oryza sativa* L., family: Poaceae or Gramineae) is the most cultivated cereal globally (after wheat) and a staple food crop for billions of people living in developing countries. This crop is grown under varied environments, over a wide range of latitudes, altitudes, and topographies with varying hydrology. Rice, mostly associated with ample quantities of water, has very low water productivity, in the sense that, to produce 1 kg of grain, it consumes 3000–5000 L of water, and, out of the 70–80% of freshwater diverted toward agriculture, rice accounts for 85% of this portion globally [1]. According to predictions, roughly 39 million ha of irrigated rice might face economic or physical water scarcity in Asia alone by 2025 [2]. The upcoming decades would be critical for transforming current agricultural practices into a more holistic approach, wherein environmental health and socioeconomic liabilities are accounted for. Therefore, it would be wise to shift from conventional transplanted puddled rice (TPR) to other systems that increase water productivity without any yield penalty. Direct-seeded rice (DSR) is such a system of rice cultivation that has the potential to decrease water consumption, in addition to lessening labor requirements and at the same time enhance resource-use efficiency and system productivity, and check greenhouse gas emissions. In recent years, DSR has been accepted as the principal rice establishment method in developed countries and adopted on >25% of the area under rice cultivation worldwide [3]. It is gaining popularity in countries such as the Philippines, India, Thailand, Cambodia, Laos, and Indonesia in tropical Asia and in the United States, Australia, and Latin America [4].

Considering its benefits from sowing to harvesting, DSR as a whole saves about 50% of water and labor expenses (Figure 1) [1], provides better temporal isolation for succeeding crops, and decreases emissions of greenhouse gases [5]. Furthermore, dry-DSR could effectively use the early-season monsoon in areas with limited moisture [6]. The DSR system uses 60.3% vis-à-vis 92.4% by TPR in non-renewable energy consumption, with average energy-use efficiency of 7.3 and 4.4 for the DSR and TPR systems, respectively [7]. However, when we assess the risks involved, yield in DSR is often diminished because of intrinsic problems such as poor seedling establishment owing to waterlogging immediately after sowing, heavy weed infestation, low nutrient- and water-use efficiency, and susceptibility to lodging [8]. Water stagnation on the soil surface after seeding can lead to poor germination due to oxygen deficiency [9]. After germination, rice plants with greater seedling vigor and anaerobic germination with good crop stands are important resources in direct-seeded rice research to mitigate these problems. Another major problem arises from the high seed rate involved in this type of cultivation practice compared with that of its counterpart, TPR. The high seed rate, besides leading to a dense population, attracting more insects and pests, and inducing nutrient competition, can be the causal reason for lodging with poor root structure [10]. Root lodging is one of the most common problems, especially when seeds are surface-sown under the dry-DSR system. This phenomenon is observed when the intact culm above the crown leans to one side due to its roots’ inability to anchor aptly in the ground. As the roots develop in this type of seeding, they produce more vertical roots with very few surface roots [11], and the anchorage is not strong enough for the plants to withstand heavy wind or rain, which often causes stem bending (middle of the internode) or stem breaking (of the lower culm below the third internode) [12]. As a result, the breaking of the stem due to lodging hinders grain filling, and transport from the source (photosynthetically active leaves) to the sink (developing panicles) is impaired.

Roots are the primary organs for sensing and buffering against several abiotic stresses (drought, flood, salinity, and mineral stress). The root system spreading inside the soil is the key player for maintaining the aboveground parts, enhancing their capacity to uptake more water and nutrient from the soil environment and yield higher. However, most breeding programs have been focusing on the aboveground plant parts. This is because root traits related to high nutrient acquisition efficiency (NAE) have never been exploited as a selection criterion because of ignorance or inadequate screening methods. Unfortunately, such traits have often been subjected to neutral or even negative selection due to the practice of high-input agriculture. Designing root architecture befitting DSR has a high potential to break both the adaptability and yield barrier. This would effectively aid in developing genotypes on par/better performing under DSR conditions than under TPR conditions. As the varieties suitable for TPR conditions are not fit for DSR [13], research work should mainly identify the novel QTLs/genes associated with improving traits under DSR to design high-yielding DSR varieties. Such an initiative would require comprehensive knowledge about the roots, the molecular mechanisms of mesocotyl growth, genes/QTLs controlling root growth, the physiological function of roots (e.g., water and nutrient uptake), root plasticity, plant–soil–microbe interactions, hormonal secretions, and their interactions as well as high-throughput phenotyping techniques under DSR. The development, popularity, and adoption of DSR and varieties will be realized only if the associated hurdles are overcome by breeding suitable varieties with high crop stand under less water, adequate nutrient availability, high competitive ability against weed infestation, and overcoming other related hindrances. Keeping all these facts in mind, this review aims to expound the understanding and updated information on root system architecture (RSA), gene networks, associated QTLs underlying root traits, and ways to exploit these QTLs for RSA to better understand water and nutrient acquisition. In addition, we suggested a model for an ideal RSA with necessary genes/QTLs suitable for the DSR system.

## 2. Insights into Rice Root System Architecture

Roots offer an interface between plants and the complex environment of soil. The root system architecture is a combined perception and spatial organization of root systems that sums up various aspects of root structure and shape. RSA is regulated by many factors such as genetics, edaphic conditions, planting density, plant size, intercropping patterns, agronomic practices, and seasonal weather patterns. Being a monocot, the rice plant bears the fibrous root system mainly of three distinct types of roots; embryonic roots, seminal roots, and post-embryonic roots. The radicle is the first root to be formed from the embryo and, after the seed germinates, the radicle is referred to as the seminal root. The parenchyma cells at the base of the stem give rise to shoot-borne roots called the crown roots. The lateral roots are borne initially on the seminal roots, and then the crown roots contribute to water and nutrient uptake activity [14]. Set apart from these structures are root hairs, fine tubular outgrowths of the epidermal layers. They are of prime importance in increasing the surface area and reaching out to the minute pore space in the soil to acquire moisture and nutrients (they also play a role in interaction with soil fauna) [15]. A plethora of natural diversity in rice RSA has been reported by several researchers [16,17]. Courtois et al. [18] identified 29 root parameters from a group of mapping populations and summarized 675 rice QTLs associated with RSA. As a consequence of the diverse, composite nature of roots and the multiplicity of functions, research work on RSA has become a collaborative field that includes communities from ecologists, geneticists, molecular biologists, and crop physiologists to microbiologists. Several genes for root development have been isolated in rice by studying abnormal root mutants [19,20]. However, unlike other traits in rice plants, the genetic bases of RSA are not entirely understood as they are complex and controlled by several genes and difficult to phenotype [21]. A list of root-related QTLs and their position are presented in Appendix A.

The modern agricultural system faces significant challenges in improving crop plants’ nutrient acquisition under dynamic environmental conditions. Thus, genetic improvement in RSA can be regarded as a prerequisite for enhanced nutrient acquisition from water stagnant to aerobic conditions. Designing a new root ideotype adapted to diverse environmental factors requires amelioration of ideotype breeding with root trait QTLs through marker-assisted selection. Therefore, it is essential to have an updated understanding of the genetic mechanism associated, the gene networks involved, and plant requirements at different stages of growth [22]. Introgression of specific RSA-associated QTLs will provide genetic progress for resistance against several abiotic stresses and the ability to grow under variable environments with a higher yield. Such research requires three steps for engineering RSA: cloning the QTLs and developing a platform of functional sequences, developing a standardized method of phenotyping, and augmenting “omics” approaches and plant breeding, biotechnology, plant physiology, agronomy, and other related disciplines to develop enhanced RSA.

## 3. RSA: Infant to Early Vegetative Stage and Associated Traits

In a well-drained upland soil, the coleorhiza emerges first from the rice seed, whereas, in submerged conditions, the coleoptile arises prior to the coleorhiza. The first roots to come out of the seed are the embryonic roots (or radicle), which arise out of the coleorhiza [23]. Then, the secondary roots develop, which ultimately form the lateral roots. The embryonic roots by this stage are desiccated and are replaced by the secondary adventitious roots (or crown roots), generating from the meristematic cells of the culm’s underground nodes. The crown roots are of two types: lower thick ones and thin upper ones. Their growth angle decides the fate of the distribution of the whole root system in the soil in the DSR system [24]. Short roots develop into a compact root system, whereas long roots help mine water and nutrients from a depth. A schematic representation of genes required for root development at different stages of growth and the ideal nutrient-specific transporters are shown in Figure 2.

### 3.1. Root Requirements and Nutrient Uptake

The desirable architecture at the initial stage of growth after germination is preferably a shallow and well-spread surface–subsurface root system on the nutrient-use front. Studies have confirmed the accumulation of phosphorus (P), one of the essential macronutrients, in the shallow soil layers vis-à-vis subsurface/deeper layers [25], and the same applies to other immobile nutrients such as potassium (K), manganese (Mn), and iron (Fe). P uptake starts as early as 2 days after germination (DAG) [26]. It is significant to note here that this uptake and activation of P transporter genes are independent of the seed-P reserve. Primarily, *OsPT8* and *OsPT1* genes are involved at the initial stage and contribute to P uptake by roots as early as 2–3 DAG [27,28]. Phosphorus starvation tolerance 1 (*PSTOL1*), a significant gene identified in the phosphorus-deficiency tolerance QTL, *phosphorus uptake 1* (*Pup1*), aids in the uptake of P and enhances early root growth [29]. It helps the plant in P uptake under rainfed/upland conditions in rice [30] and was reported in an *aus*-type variety Kasalath [29]. Additionally, Sandhu et al. [31] have identified QTLs regulating P uptake and length of root hair on chromosome 5. The key player in developing different types of the root system in the early seedling growth are the genes controlling root angle, which determine the development of surface and shallow subsurface roots. However, root growth angle as a trait is determined by multiple environmental factors such as gravity, light, and water potential [32,33]. An actin-binding protein, Rice Morphology Determinant (RMD), has been found to govern root angle by linking actin filaments with statoliths producing shallower crown root angle [34]. This also leads to a low intensity of auxin-driven gravitropism of the root. Another QTL, *qSOR1* (QTL for SOIL SURFACE ROOTING 1), an 812-kb long segment located on chromosome 7 expressed in the columella cells, also operates similarly working antagonistically with auxin [35]. It induces shallow root formation beneficial for the uptake of nutrients from the top layers [36]. *DEEPER ROOTING 1* (*DRO1*), a major QTL responsible for root growth angle, is also involved in gravitropism [37] and is negatively regulated by auxin signaling and is involved in root tip cell elongation leading to gravitropic bending [38]. Apart from the genes mentioned above, many other QTLs for root growth angle have been reported in rice, such as *DRO2* [38], *DRO3* [39], *DRO4*, and *DRO5* [40] detected in F_2_ mapping populations derived from a cross using “Kinandang Patong” as a large root growth angle donor. Different flanking QTLs have been reported from chromosomes 2, 4, and 7 near *DRO4*, *DRO2*, and *qSOR1*, respectively [41], which can be used further. Interestingly, it is seen that the plants with shallow rooting capture Cd from the top soil layer; thus, the shallow rooting allele is a potential genetic resource for phytoremediation under high Cd accumulation. When considering eating purposes, the allele for deep rooting could be useful to avoid absorbing bioavailable Cd in such soil conditions [33]. Besides the above, the crown root has an important role in P acquisition. A large number of crown roots (adventitious roots) contribute to shallower rooting depth, in turn increasing P uptake from deficient soils [42]. Enhancing P uptake by introgression of these QTLs and genes can alleviate the problem of P deficiency mainly prevalent in dry conditions of DSR (Appendix A).

Nitrogen (N) deficiency at the early stages of seedling growth can lead to the development of long but thinner roots because starvation increases the rate of cell division and cell elongation, thereby increasing the length of primary roots [43]. This shows the plant’s preference for root growth as against shoot growth in such situations. Rice roots can uptake N in both the nitrate and ammonium form. Under limited-moisture DSR conditions, the availability of nitrate would be higher than the ammoniacal form of N. However, heavy rainfall or flooding in rainfed lowland would change the soil environment. Therefore, roots should have plasticity for this situation, and the presence of both nitrate and ammoniacal forms of transporters is highly required. At the young stage, specific localized nitrate signaling by the root helps develop lateral roots that decide the root branching system [44]. Nitrate uptake has been reported to be facilitated by transporters such as the *OsNRT1.1* allele *OsNPF6.5* (*OsNRT1.1B*), *OsNPF2.4* (*OsNRT1.6*), *OsNPF8.9* (*OsNRT1.1*, *Os3g13274*, *or AF140606*), and *OsNRT1.1b* (AK066920) in rice [45,46,47] under high nitrate availability, whereas, in a soil environment with low nitrate, *OsNRT2.1*, *OsNRT2.2*, and *OsNAR2.1* come into play [48,49,50]. The partner protein of the NRT family, *OsNAR2.1*, also helps in promoting root growth. Interestingly, one of these transporters (*OsNRT1.1B*) is responsible for the wide variation in nitrogen-use efficiency (NUE) observed between *indica* and *japonica* subspecies of rice (*Oryza sativa*) [45]. Under oxygen-limiting conditions, rice can also rely on ammonia for its N source. This uptake in the proper amount is carefully controlled by ammonium (NH_4_^+^) transporters (AMTs) [51]. A range of transporters for ammonium intake by the plant cells have been reported such as *OsAMT1.1*, *OsAMT1.2*, *OsAMT1.3*, *OsAMT2.1*, *OsAMT2.2*, *OsAMT2.3*, *OsAMT3.1*, *OsAMT3.2*, and *OsAMT3.3* [52,53,54]. A study revealed that deep rooting gene *DRO1* enhanced N uptake, as longer roots could mine nitrate ions from the lower layers of the soil [33]. They compared IR64 and *Dro1*-*NIL* for N uptake and found that the latter had higher uptake with higher grain filling. A major QTL on chromosome 12, *Tolerance of Nitrogen Deficiency 1* (*TOND1*) contributes to N deficiency tolerance in plants, also manifesting increased root length [55]. QTLs for N uptake and root hair density have been identified on chromosome 2 (a stretch of 3.4 Mb) [31]. Yet another QTL, *NITRATE TRANSPORTER 1.1*, has been reported, which transports both nitrogen and auxin, and this was one of the first studies to suggest a connecting link between the auxin pathway and nitrate availability in soil [56]. In this context, it is noteworthy to mention that, under N starvation, many phytohormone activities also become regulated, such as Indole acetic acid (IAA), Abscisic acid (ABA), and Jasmonic Acid (JA) [57].

Besides N and P, iron (Fe) can greatly influence and limit plant growth. The peculiar case of this nutrient is that its availability cannot be compensated by external application. Even when it is present in the soil, it remains in unavailable form because of its solubility. The aerobic conditions under DSR lead to oxidation of iron from its available ferrous (Fe^2+^) form to its unavailable ferric (Fe^3+^) form [58]. Under normal conditions, rice, being a member of the *Poaceae* family, proceeds through the biosynthesis and secretion of root exudates, compounds called phytosiderophores (PSs), through the roots to facilitate the uptake of Fe. They are produced in response to the levels of different enzymes such as nicotianamine synthase (NAS), nicotianamine aminotransferase (NAAT), and deoxymugineic acid synthase (DMAS) [59]. These PSs form soluble complexes with iron (Fe(III)-PS) in the rhizosphere and then are taken up by the root cells with the help of *YELLOW STRIPE-LIKE PROTEINS* (*YSLs*) [60]. Rice plants, especially at the early stage, are highly susceptible to Fe deficiency as they secrete a lower amount of deoxy-MA even under Fe-deficient conditions, and the secretion completely stops within 7 days [61]. When iron depletion continues, the rice root tips become chimeric, whereas the epidermal cells become necrotic. Besides this, rice acquires Fe^2+^ directly from the surrounding rhizosphere through multiple transporters such as *OsIRT1* and *OsIRT2* [62].

Zinc (Zn) is one of the most critical regulatory cofactors and a structural component of several biomolecules involved in various biochemical pathways. Its deficiency can cause multiple irregularities, including chlorosis and hampering auxin and chlorophyll activity. Plants tolerant to Zn deficiency in soil show rapid crown root development [63]. Zinc uptake by plants is facilitated by one of two ways: directly from the soil via root epidermal cells or via associations with arbuscular mycorrhizal fungi (AMF). The direct uptake from the soil is similar to that of iron. Like iron, response to Zn deficiency is related to root exudates, phytosiderophores (MA). *ZmYS1*, a yellow stripe gene, identified in maize, has a range of specificity for metals, including Zn, and thus helps in uptake and transport [64]. Many Zn transporters of the ZIP (*Zrt*, *Irt*-like protein) family have been reported that sense and become activated in response to Zn deficiency: *OsZIP1*, *OsZIP3*, *OsZIP4*, and *OsZIP5* [62,65,66]. These *ZIP* transporters are also involved in the transport of iron and manganese [65]. Besides root exudation, zinc uptake is facilitated by mycorrhizal fungi (as is phosphorus). These fungi colonize the root cortical cells, extending their hyphae into the rhizosphere [66]. This helps to supplement the existing surface area of the roots, thereby exploring larger parts of the soil and facilitating increased uptake. Such a symbiotic relationship can be more beneficial, especially when the nutrient in question is an immobile one, such as zinc or phosphorus [67]. Much has been studied and explored about AMF symbiosis in numerous crop plants, but rice AMF symbiosis is yet to be exploited. According to reports, there was an increase of 28–57% in root colonization when inoculation occurs in aerobic rice, owing to the non-flooded conditions [68]. This report on AMF associations sheds light further on new dimensions to be exploited under dry-DSR system RSA.

### 3.2. Early Uniform Emergence and Early Root Vigor

Early uniform emergence (EUE), seedling establishment, and development are major determinants of crop growth and subsequent yield. EUE prefers a deep root system that acclimatizes the emerging seedlings well before the upper soil becomes dried. This is contributed by several traits: early and high germination rate, rapid root-shoot development, and enhanced seedling vigor. Two significant major QTLs (*qEMM_1.1_* on chromosome 1 and *qEMM_11.1_* on chromosome 11) for EUE derived from Moroberekan (BC_2_F_3_; Moroberekan × Swarna) were reported by Dixit et al. [69]. QTLs for early uniform emergence (*qEUE11.1*) [69] and early vigor (*qEVV9.1*) [70] have been reported that can be used for the improvement of dry-DSR varieties. A QTL for the seedling establishment was reported on chromosome 11 under DSR along with grain yield and its contributing traits [31]. These QTLs can be utilized to assist the early vigor of the seedling under DSR. Early vegetative vigor (EVV) or seedling vigor is the plant’s ability to emerge rapidly from the soil, establish a viable seedling, and acquire a high relative growth rate (RGR) before canopy closure [71,72]. Seedling vigor traits would include uniform germination, longer coleoptile, and mesocotyl accompanied by rapid growth of both root and shoot [73,74], which further contribute towards optimum DSR crop establishment and at the same time competitiveness against the growth of weeds [75,76]. Mesocotyl, the portion between the coleoptile node and the base of the seminal root, is one of the key players in pushing the germinating seedlings out of the soil. Its length determines the seedling emergence and establishment. GWAS study for mescotyl elongation in 208 accessions of rice has identified six novel loci explaining up to 15.9% of phenotypic variations [77]. Three QTLs, namely *qMel-1*, *qMel-3*, and *qMel-6* controlling mesocotyl length, have been reported on chromosomes 1,3 and 6 respectively in backcross inbred lines developed from a cross between Kasalath and Nipponbare [78]. The significant differences between rice cultivars under direct-seeded rice have been well established owing to the variation in their EVV [76,79]. In the DSR environment, EVV is associated with yield stability [80]. Several researchers [73,76,81,82,83] have already identified many QTLs for these traits using mapping populations. Two major QTL hotspots have been identified in relation to vigor traits: QTL hotspot A on chromosome 3 (*qEV3.1*, *qEUE3.1*, *qSHL3.1*, *qSL3.1*, *qSFW3.1*, *qTFW3.1*, *qRDW3.1*) and hotspot B on chromosome 5 (*qEV5.1*, *qEUE5.1*, *qSHL5.1*, *qSL5.1*, *qSFW5.1*, *qSDW5**.1*, *qTDW5.1*) [84]. Zhou et al. [85] have discovered two rather more stable QTLs (*qFV-5-1* and *qFV-10*) that contributed to vigor under both high and low water stress.

Another aspect of seedling vigor is the proper and early development of crown roots, a major trait in rice that controls nutrient and water acquisition in the initial days of seedling growth. Thus, their number and distribution are deciding factors of ESV, and they also help in P uptake due to their shallow rooting. On the contrary, it has been reported that in maize, a lower number of crown roots facilitated enhanced nitrogen acquisition from soil [86]. A QTL for nodal roots was identified on chromosome 4, in a stretch of 760 kbp [31]. Rapid nodal root growth, longer root, and high density of roots are imperative for EUE under DSR [31]. For their timely normal emergence from the shoot base, seedlings with high vigor have been reported to utilize the WUSCHEL-related homeobox (WOX) gene, *WOX11* [87]. They found its action to be in coordination with both auxin and cytokinin activity. A mutant identified in rice, *crown rootless1* (*crl1*), showed decreased lateral root formation, abnormal crown root formation, and compromised root gravitropism. The expression of *crl1* leads to normal root morphology. Exogenous application of IAA was also effective in inducing the expression of the DR5 promoter in relation to the normal development of crown roots [88]. *Oscand1*, a gene reported by Wang et al. [89] whose mutant develops a defective crown root system, is due to inhibition at the cell division’s G2/M transition phase. The peculiarity is that this gene also has auxin signaling in its functionality [89]. In this context, one must also focus on the development of lateral roots. Both crown root (CR) and lateral root (LR) developmental processes follow many common pathways and gene actions. For instance, the WOX genes, *OsIAA23*, *OsARM1*, *OsARM2*, and *CRL4*/*OsGNOM1*, have a regulatory function in both LR and CR development [90]. It is interesting to note that crown root regulating significant gene, *crown rootless1* gene also positively regulates the development of LRs [22]. LRs are one of the major contributing traits to root architecture as they contribute to root biomass and also facilitate better anchorage. However, not much significant work has yet been reported regarding the expression and regulation of LR-controlling genes. A recent report on *lateral rootless 1* (*lrt1*) and *lrt2* mutants suggested that they had less sensitivity to auxin and lacked LRs [91,92]. Similar auxin involvement is also seen in the *OsWOX3A*-encoding genes *NARROW LEAF 2* (*NAL2*) and *NAL3*, which regulate LR development. *NAL1* has also been identified with a role in crown root development [93]. Contrary to these facts, EVV may also sometimes hamper normal crop growth as the early vigorous uptake of nutrients, and moisture leaves very little for the crop at later stages of DSR when resources are limited [94]. A study on maize suggested that nitrate uptake is optimized by long and sparse lateral roots, whereas P uptake is optimized by short and dense lateral roots [95]. Thus, we need to incorporate both genes governing RSA into a single genotype for optimum utilization of N and P.

## 4. Stress during Germination

Roots face several stresses, both during and after germination. It is well understood that the early vegetative/seedling stage exposed to drought, heat, osmotic stress, and nutrient deficiency leads to alteration of the plant’s root-shoot ratio as an adaptive strategy. At the initial stage, the belowground organ (root) sends signals to the aboveground portion and alters both the root and shoot architecture. Roots inherently possess adaptive mechanisms to cope with such stress by expressing stress-responsive protein and biochemical pathways. The following sections will discuss such stress conditions.

### 4.1. Anaerobic Germination and Regulation of Associated Root Traits

Water, when available in excess, can be a limiting factor in the early stages, hampering the pathway of germination. This is one of the most common stresses that a germinating seed faces, i.e., anaerobic soil conditions. Even though DSR conditions ideally should be aerobic, at the beginning of seed germination, surplus water stagnation causes major germination failure. On the contrary, this can aid in weed control during the initial stages. Few rice cultivars have the ability to germinate, grow, and survive under such oxygen-deficient conditions is commonly known as anaerobic germination (AG) tolerance in the crop. Such a condition of low oxygen can sometimes lead to hypoxia. Generally, lowland rice cultivars can germinate without oxygen as they can synthesize the enzymes required for starch degradation even in oxygen-depleted soil layers [96]. However, when we breed for DSR cultivars, we cannot overlook the menace of anaerobic conditions prevailing in this system. The *Os-EXP4* expansin gene helps in seed germination in anaerobic conditions or submergence. This gene might be involved in the expansion of the epidermal cells to form long coleorhizal hairs [97]. Among the several QTLs detected for anaerobic germination, *qAG-9-2* was traced to the locus *AG1* (trehalose-6-phosphate phosphatase gene family) controlling coleoptile elongation under submergence [98]. A similar coleoptile elongating gene (*AG2*) has also been reported. These genes (*AG1* and *AG2*) have displayed surprisingly higher survivability under anaerobic conditions in the introgressed lines than in the recurrent parent, Dongan, a *japonica* cultivar [99]. However, *SOR1* (*SOIL SURFACE ROOTING 1*) remains the most useful gene adapted to anaerobic conditions under DSR. Its mechanism enables roots to grow toward the soil surface and acquire oxygen directly from the air [100]. Uga et al. [101] identified an 812-kb interval (7L), delimited by markers RM21941 and RM21976, and it was designated as QTL *qSOR1*. This SOR gene is reported explicitly in the Bulu genotype from Indonesia. The evolution of this gene might be attributed to the selection pressure to withstand anaerobic growth conditions. Unlike lowland ecotypes that develop thinner superficial roots after the panicle initiation stage, Bulu ecotypes develop thick crown roots above the soil surface beginning at the seedling stage. A similar mechanism is adopted by *Arabidopsis* using an allele of *CYTOKININ OXIDASE 2* (*CKX2*), which promotes shallower root growth aiding in surviving hypoxia due to snow [102]. Exploiting such QTLs can help overcome hypoxia and anaerobic germination and also aid in the uptake of P. A recent report on identifying underlying QTLs for anaerobic germination [103] highlighted major QTLs on chromosomes 3, 5, 6, 7, and 8 in two mapping populations using a common parent (Kalarata) with the anaerobic germination trait. Of the five QTLs identified in the study, *qSUR6-1* was a novel one for anaerobic germination. The Korean weedy rice, photoblastic rice (PBR), was identified with potential for anaerobic condition survivability with high germination percentage [104]. Jeong et al. [105] developed a mapping population using PBR, and the authors reported three QTLs involved in imparting tolerance of flooding: *qAG1*, *qAG3*, and *qAG11*. Other genes have also been reported, such as the QTLs *qAG-9–2* (*AG1*) and *qAG-7–2* (*AG2*) [106]. QTLs located on the short arm of chromosome 7, *qAG7.1* and *qAG7.2* [107], and the functional allele of *qLTG3-1* in Ouu 365 and Arroz da Terra inbred lines [108] were found to promote efficient germination under anaerobic conditions. The National Rice Research Institute (NRRI), Cuttack, in collaboration with the International Rice Research Institute (IRRI), has started several breeding programs for the introgression of reported QTLs for anaerobic germination (*qAG9.1*, *qAG9.2*) to develop high-yielding DSR varieties [109].

### 4.2. Limited Moisture during Germination and Regulation of Associated Root Traits under DSR

Approximately half of the world’s rice production is dependent on rainwater and is grown in aerobic upland or rainfed lowland systems, and the plants are often exposed to unmitigated drought stress [110]. Breeding approaches for high-input agriculture have favored a shallow root system that absorbs nutrients from the top layer of the soil. On the contrary, plants bred to sustain themselves in a low-input rainfed ecosystem require a robust and deeper root system to acquire most of the water and nutrients to achieve their potential [111]. In plants, roots are the primary organs that sense moisture stress and initiate a signaling cascade at the molecular level. It has been seen that rice roots respond to drought or water stress in the following ways: osmotic adjustment within the root cells, enhanced root penetration into the soil by increasing root length, increased root density, and a higher root-to-shoot ratio [112]. These responses are always determined by the plants’ genotype, intensity, and period of exposure to the stress. Drought affects rice production in three common ways: early water stress (causes a delay in transplanting of seedlings), mild sporadic stress (having cumulative effects), and late stress (near to flowering) [113]. However, the basic adaptive response in the DSR system lies with traits such as root length, root thickness, and root hair growth to reach moisture at greater depths. Root elongation is facilitated by auxin signaling and the expansion of the cells. Several factors control root length as a trait, for instance, cell wall loosening regulated by the endo-1,4-β-d-glucanase protein encoded by the gene *ROOT GROWTH INHIBITING* (*RT*)/*OsGLU3* regulates root length [114]. Kitomi et al. [115] identified two QTLs for maximal root length, *QUICK ROOTING 1* (*QRO1*) on chromosome 2 and *QRO2* on chromosome 6. The *rt*/*osglu3* mutants exhibit short roots as a decrease in longitudinal cell elongation occurs but with no effect on root differentiation, root cell division, or shoot development. *OsEXPA8*, a turgor-driven cell elongation root-specific α-expansin, works similarly by loosening the cell wall and then resulting in an increased seminal, crown, and lateral root length [116]. An extensive meta-QTL analysis across populations and environments, reported by Courtois et al. [18], has shed light on 119 QTLs distributed among hotspots of chromosomes 1 and 9. Most of them were related to root length. Many other genes controlling root elongation have also been found to be effective, such as *GNA1*, which encodes a glucosamine-6-P acetyltransferase; *OsCYT-INV1* (an alkaline/neutral invertase) [19]; *Osglu3-1* (a putative membrane-bound endo-1,4-beta-glucanase) [117]; *OsRPK1* (a Ca^2+^-independent Ser/Thr kinase [118]; QTL *Dro1* (*DEEP ROOTING 1*) for deep rooting (with an increased gravitropic response) [101]; and *O. sativa CELLULOSE SYNTHASE-LIKE D1* (*OsCSLD1*) [119]. Wang et al. [120] identified the QTL *qRL7* (657-kb interval on chromosome 7), and Qu et al. [121] identified *qRL8* that regulates root length at later stages of plant growth, specifically the heading stage. Besides root length, upland cultivars with a thicker coarse root system and with high root length density respond better to water stress [122]. Root thickness determines the water uptake, nutrient acquisition, and penetration of the roots [91,123]. QTL *qRT9* that has a function regulating both root length and root thickness has been identified. It encodes a basic helix–loop–helix (*bHLH*) transcription factor, *OsbHLH120*. Its level of expression is in turn steered by drought-response phytochemicals such as salt, polyethylene glycol, and ABA [124]. In addition to these traits, increased root hair can increase the surface area and enhance moisture and nutrient acquisition from the finest soil pores that remain inaccessible to the root apex [125]. Like root length and thickness, root hair is also linked to a *bHLH* transcription factor encoded by *O. sativa ROOT HAIRLESS 1* (*OsRHL1*) controlling cell epidermal transformation [126].

Besides water stress, in optimal growth conditions, root hair growth is triggered by several factors that include cytoskeleton restructuring, cell wall loosening (as also seen for root elongation), calcium ion concentration, pH of the cells, and auxin and ROS levels (accumulated under stress) [127]. Two SNPs, S5_15470847 and S5_15470880, identified for root length and root density associated with the genes *OsIPT3* [128] and *OsEXPA3* [129] were also found to be active in the regulation of the length of vascular bundle cells in the root, length of the primary root, and its density. As mentioned earlier, expansin genes play an important role in most root traits. It is obvious that whole root system architectural development is an interrelated and correlated manner of growth. Root length-controlling expansin genes *OsEXPA17*, *OsEXPA30*, and *OsEXPB5* all conserved domains for root hair-specific elements (*RHEs*) tightly linked with root hair initiation [130,131]. Reduced or short root hair has been observed in mutants of *O. sativa SEC14-NODULIN DOMAIN PROTEIN* (*OsSNDP1*) that encodes a phosphatidylinositol transfer protein [132] and short root hair 2 (*srh2*) mutant, with a mutation in the *XYLOGLUCAN XYLOSYLTRANSFERASE 1* (*OsXXT1*) gene [133]. Root pulling resistance, a trait that has a high positive correlation with root length, root thickness, branching number, and dry mass in rice [134], is also an indication of drought tolerance (possessing a larger root system).

Different intensities of drought induce different responses in plasticity. Plasticity in the roots can range from root length density [135,136] to lateral root length and/or branching [137,138,139], influencing a variety of traits such as shoot biomass, water uptake, and photosynthesis under moisture limited condition in rice. The plasticity in the development of arenchymatous cells and lateral root growth [140] resulted in higher grain yield [141] as well as shoot dry matter [142] when the plants were subjected to transient drought stress. Two loci (id1023892 and id1024972) tracked to a region near *qDTY_1.1_*, a major-effect drought-yield-related QTL on chromosome 1, provided plastic responses to drought by regulating root and shoot growth [143] and also enhanced the level of deep roots in the OryzaSNP panel [144]. Apart from the above, the recent reports on the role of aerenchyma tissue that aids tolerance to moisture stress. The formation of lysogenous cortical aerenchyma cells helps the plant to reduce the metabolic cost and allowing for greater water uptake from soil [145]. As DSR conditions are mostly aerobic systems, developing a drought-tolerant root architecture expressing the required genes and morphological traits is essential.

### 4.3. Cold Stress and Salt Stress during Germination

Low temperatures at the seedling stage can diminish yield by inhibiting germination and photosynthesis, retard growth, and slow down different biochemical pathways [146]. Apart from this, low-temperature stress can lead to the accumulation of cell membrane-damaging entities such as reactive oxygen species (ROS) (singlet oxygen, superoxide anions, hydrogen peroxide, etc.) leading to electrolyte leakage and lipid peroxidation [147]. Cold stress most commonly decreases the hydraulic conductivity of the roots, thereby decreasing the supply of nutrients to the shoots [148]. During cold stress, a multifold increase is seen in root aquaporin gene expression (*OsPIP2.5*) as a result of a shoot-to-root signal [149]. There seems to be a relationship between decreasing hydraulic conductivity and increasing aquaporin activity of roots under cold stress, especially in rice [150]. Several aquaporins belonging to intrinsic protein families have been reported to regulate hydraulic conductivity in cold stress [151], and that needs to be exploited further.

The problem of salinity or salt stress has been increasing gradually, resulting mostly from faulty agricultural practices (frequent light irrigation, higher use of fertilizers such as muriate of potash and ammonium sulphate and summer fallowing) [152,153,154]. The damage caused by salt stress starts right from seed germination; lower water uptake, decreased rate of photosynthesis, and consequently oxidative stress [155], which is ultimately responsible for decreased growth. Similar to cold stress, plants under salt stress undergo oxidative stress and produce peroxidases leading to cell wall stiffening [156]. A general study for screening tolerant genotypes for these stresses (both cold and salt stress) would be to look for plants producing oxygen-scavenging elements [157], detoxifying ROS, and methylglyoxal (MG). Another detrimental effect of a saline soil environment is that the higher concentration of sodium ions (Na^+^) near the root zone antagonistically inhibits potassium ion (K^+^) uptake by the roots [158]. As K^+^ is responsible for maintaining membrane potential and turgor pressure inside cells, its absence leads to impaired cellular activity. Besides K^+^, a decrease in the uptake of P and Zn occurs. Additionally, because of an increased accumulation of Na^+^ and Cl^−^inside the cell, several biochemical processes such as the synthesis of biomolecules such as proteins and their enzymatic activities are deregulated [159]. Several genes have been reported for salt tolerance with respect to shoot growth, but little work has been reported on the root front. However, from the root perspective, the desirable trait would be to form a hydrophobic barrier; for instance, suberin lamellae (SL) and Casparian bands (CBs) are hydrophobic barriers in roots that block the apoplastic leakage of water and ions into the xylem in various plants [160].

## 5. RSA: Vegetative to Reproductive Stage

The seedling stage is delimited by the appearance of the first tiller and then the vegetative growth stage starts. When the plant reaches the vegetative stage, ground cover occurs faster due to EVV, and this lessens soil evaporation and weed growth and accelerates root uptake of soil water and nutrients. In terms of higher acquisition of nutrients and water, the roots at this stage should preferably be deep. This can increase uptake from deeper soil layers (as topsoil and subsoil layers must have already been exhausted by now) and help overcome drought or water stress at the later stages of plant growth. It was seen that premature leaf senescence occurred due to lower root length and root number, which made N availability limited in DSR conditions [161]. Generally, under DSR conditions, the roots are shallow, but, if this trait is improved, then lodging can be controlled. During the grain-filling stage, N application can lead to an increase in root length and root area. This enhancement can maintain active root activity for a more extended period. Consequently, leaf senescence can be delayed and the active photosynthesis period prolonged, resulting in higher grain filling [162]. This would also facilitate higher water- and nutrient-use efficiency. A study using different densities of DSR sowing along with different rates of N was carried out by Deng et al. [163], who found a significant association between grain yield and total root length. Additionally, they concluded that high-density sowing of DSR along with low N application could lead to an improvement in root morphology. This enhancement in root morphological traits such as root number, length, surface area, and volume contributes to higher grain yield.

The root length, root number, root density, root thickness, and lateral branching or lateral roots are important at the later stages of plant growth. A QTL controlling linear lateral root number, L-LRN (*qLLRN-12*), on chromosome 12 guided the development of long-type LR production [140]. NRRI, Cuttack, has used QTLs for higher root length density (*qNR5.1*, *qRHD1.1*) and nodal root (*qNR5.1*) as donors for transfer of these specific traits to improve genotypes of dry-DSR [109]. Root volume also has a role for auxin in it, and a candidate gene (*NAL1*) regulates both leaf and root growth [164]. This gene is associated with the QTL *qFSR4*, which has a role in root volume per tiller, with a 38-kb segment fine mapped on chromosome 4 [165]. LR regulation and root thickness have already been reviewed in previous sections. Another important root organ that not many have focused on is the root tip. According to Robinson et al. [166], continued growth and production of root tips for mobile resource uptake might be more important than total root length. Because of suberisation with time and also exposure to dry soil [167], roots tend to be more apoplastic in nature. As a result, roots have a decrease in water uptake. However, if the root tips are regenerated or remain unsuberised, then this loss in water uptake can be compensated to some extent [168]. Both auxins (*OsIAA23*, *OsARF16*, *OsWOX11*, and *OsWOX12*) and cytokinin play a role in root cap regeneration [133]. Considering water and nutrient translocation, xylem and stele structures are also more important than root thickness [169]. *STELE TRANSVERSAL AREA 1* (*STA1*), a QTL on chromosome 9 (mapped in the vicinity of *DRO1*), controls the stele transversal area [169,170].

During the later stages of plant growth, demand for nutrient uptake increases vis-à-vis the seedling and early vegetative stage. A total of 59–84% of the nutrient uptake by the rice plant takes place from the tillering stage to the anthesis stage [171]. Therefore, pressure on individual roots is high for the uptake of nutrients. Thus, there is a need for the development of new roots to decrease nutrient demand per unit volume of roots. This specifically is required at the flowering and grain-filling stage to facilitate higher grain filling. Lodging tolerance should be another major objective when breeding DSR. It is determined by various contributing traits such as medium plant height with large stem diameter and thick stem walls with high lignin content [172,173]. Root lodging-tolerant genotypes exhibited higher root volume and increased anchorage at the full heading stage than susceptible ones [174]. Additionally, strengthening the culm and lower positioning of panicles can aid in decreasing the risks of root lodging. The QTL *STRONG CULM3* (*SCM3*) develops culm strength in rice [175]. QTLs for higher nutrient uptake with lodging tolerance (*qLDG_3.1_*, *qLDG_4.1_*) are already reported [69], and a multi-QTL stacking program is being undertaken at NRRI, Cuttack, to incorporate them in breeding DSR varieties [109]. In addition, the accumulation of starch enhances the culm’s flexibility [176] and higher silicon deposition provides strength against lodging [177]. Kashiwagi and Ishimaru reported a locus (*pr15*) that enhanced the weight of the lower stem, thus displaying tolerance against lodging [178]. However, lodging can be effectively controlled only when the plant has a wider root plate. Root plate refers to the section where maximum stiff portions of the root terminate [179]. Failure in anchorage or loss in anchorage strength results in plant lodging [180]. This is in turn dependent on the plant’s root plate spread and structural rooting depth. The genetic improvement of these traits would assist in breeding cultivars tolerant of root lodging [181]. Long et al. [182] have reported several QTLs for traits that aid in tolerance of lodging. They have identified a major QTL, q*LR1* (~80 kb), that increases stem length diameter and breaking strength, *qLR8* (~120 kb) that improves breaking strength, and many others using sequencing through SNP markers. Sandhu et al. [183] have identified QTLs for traits contributing toward lodging tolerance: stem diameter (187 Mb) and bending strength (0.038 kb) on chromosome 3 and culm diameter (1.3 Mb) on chromosome 2. A strong association was observed between anchorage strength and lodging, that is, cultivars that had greater root depth and root plate spread were less prone to lodging and gave comparatively higher yield [184]. It is noteworthy to mention that deep sowing in direct-seeded rice has also become a phenomenon wherein it helps the plant to have better anchorage and imparts lodging tolerance. This would require the higher expression of mesocotyl and coleoptile elongation genes [78]. At the same time, the deep setting of seeds also helps in deeper roots and access to soil moisture from zones that otherwise would be inaccessible by the plant.

The grain-filling stage would require more carbon sources diverted to the grains from different parts of the plant. Especially in drought conditions, root carbon can act as a source for grain assimilates [185]. Thus, a higher carbon source accumulation in the roots could be a desirable trait during the grain-filling stage. Additionally, this stage requires continuous uptake of nutrients, so longer roots to acquire more N and enhance the active photosynthesis period of the leaves are more desirable [186]. A significant positive correlation was seen between the nodal roots, root depth, root hair length, grain yield, and the amount of nutrients taken up by the plant [31]. Thus, at this growth stage, a higher nutrient uptake with ample new roots would facilitate proper grain filling without compromising grain yield. The development of a few new roots can help in meeting the high nutrient demand in the resource-poor conditions of a DSR system.

In addition to this discussion on root morphologies, a few other root traits need a brief discussion. Young root tips play a major role in moisture acquisition and are regulated by different root attributes like root length and root surface area [187]. Thus, new root tips for continuous growth may be vital for the uptake of water and nutrients. Another fine organ in high proximity to the resources available in the rhizosphere is the root hair. Root hairs increase the contact area of roots with soil particles and thereby aid in the uptake of water and nutrients, and they sense biotic and abiotic stresses [188,189]. However, a few reports also suggest that root hairs may not have much to contribute toward water uptake, especially in the case of rice [190]. A decrease in root diameter and increase in specific root length lead to an increase in root surface area, thereby increasing the hydraulic conductance and decreasing the apoplastic barrier of the roots [191]. Hydraulic conductivity is influenced by the diameter of xylem vessels, and it ultimately determines plant productivity under drought stress. A lower xylem diameter will lead to a reduced hydraulic conductivity having a minimum risk of cavitation because of more conservative water use in comparison to that of the plants with higher xylem vessel diameter [192], with some exceptions [193]. Decreasing root xylem diameter through effective breeding strategies will cause a decrease in hydraulic conductance under sufficient moisture availability. These reports show that there has been a realization of root traits and their potential to increase moisture and nutrient uptake. In Australia, wheat varieties were developed with conservative hydraulic architecture in seminal roots to save soil moisture during critical crop growth stages under drought. Other root morphological traits influencing resource acquisition are increasing the number of fine roots and the rate of overall root growth.

## 6. Biotic Stress in the Form of Weeds

Among the other factors in the dry-DSR system of cultivation, weeds are the most difficult to control and, even when controlled, are a recurring menace throughout plant life. This discourages most farmers from adopting the DSR method. Even though herbicide and weedicide are easily available, the breeder community must focus on looking for an inherent genetic factor in the crop. Reports exist of efficient allelopathic rice cultivars that secrete a sufficient amount of allelochemicals to suppress weed growth in the manner of a sui generis weed management system by the rice plant. AfricaRice researchers designed a rice plant with favorable competitive traits: a cross-species hybridization between *O. glaberrima* and *O. sativa* was made to achieve the competitive ability of the former and yield quality of the latter [194]. This not only increased competitiveness but also improved yield in the derived lines. Mapping populations consisting of RILs have highlighted that the allelopathic nature in rice is a quantitatively inherited trait [195]. Two rice genotypes; Huagan-3 (commercially accepted allelopathic variety) and Liaojing-9 (non-allelopathic), were studied for their effect on paddy weeds *Cyperus difformis*, *Echinochloa crus-galli* (barnyardgrass), *Eclipta prostrata*, *Leptochloa chinesis*, and *Oryza sativa* (weedy rice) [196]. It was seen that inhibition occurred more at the root level than at the shoot level. There was a significant decrease in total root length, total root area, maximum root breadth, and maximum root depth of paddy weeds. Another remarkable achievement was the report from Chung et al. about the identification of the QTL governing the allelopathic trait [197]. The RILs generated from an allelopathic line (Sathi), and a non-allelopathic parent (Nong-an) highlighted a 194-kbp region on chromosome 8 with 31 genes located in the segment. Despite the immense work on allelopathy and its suppressing effect on weeds, the action taking place below the ground involving the roots is little understood. Apart from these facts, one important aspect of allelopathy is toxicity. Toxicity is a dosage response. Allelopathic chemicals can be as harmful as weedicides to the soil or crop residues if not checked at a threshold level. Thus, such traits must undergo rigorous scrutiny before introgression and release in plants.

## 7. Root Growth and the Role of Phytohormones

Phytohormones activate and regulate most biochemical activities throughout the plant system. They also have a huge effect on the growth and proliferation of the root system. A few of the essential functions concerning root traits are covered in the following sections. Auxin is involved in almost all root trait expressions either directly or indirectly. The inactive or quiescent center (QC) region of the root apical meristematic zone is maintained by auxin. As reported by Friml et al. [198], auxin is also responsible for root cap growth. It was observed that mutant plants with an impaired *AUXIN* (*Aux*)/*INDOLE-3-ACETIC ACID* (*IAA*) gene family, *OsIAA23*, developed damaged root caps, thereby inhibiting root growth [199]. Auxin also has a key role in radicle development: radicleless (*ral1*) mutants were defective in their response to auxin, indicating its role in radicle development [200]. Its role in crown root and lateral root formation has been highlighted in previous sections. Auxin-cytokinin ratio regulates root morphology: a high ratio favors root growth, whereas a low value or higher cytokinin level favors shoot growth [201]. Moreover, crown root development has been found to be controlled by this relation of auxin and cytokinin, which works antagonistically with each other as observed in the case of cytokinin signaling gene *WUSCHEL-RELATED HOMEBOX GENE 11* (*WOX11*). The mutant *WOX11* shows a decrease in the number of crown roots [22,87]. *YUCCA 1* (*OsYUC1*) is the key enzyme in auxin biosynthesis [202], whose overexpression enhances crown root formation [203]. Several *PIN-FORMED* (*PIN*) genes, *OsPIN10a* and *OsPIN3a* [204,205], *OsPIN2* [206], and *OsPID* [207], control the auxin efflux regulating polar transport and also help in crown root development. Mutants linked with auxin-related defects also affect the lateral roots to some extent, for example, lateral rootless 1 (*lrt1*), *lrt2*, auxin-resistant mutant 1 (*arm1*), and *arm2* [91,92]. Under low rates of nitrogen, the cytokinin signal decreases; as a result, higher root biomass is achieved [208]. The auxin–cytokinin ratio determines the development of lateral roots. However, Gao et al. [209] reported cytokinin’s (*OsKX4*) positive action in rice root development using transgenics with less than the enzyme’s regular level. Ethylene primarily inhibits root growth. This hormone either enhances or represses root growth, depending on its concentration. Mainly during drought stress, its concentration is regulated and induces different stress responses. Transcription factor *OsEIL1* promotes root elongation, which is also a component in the ethylene signaling pathway [210]. The mutant of *O. sativa ethylene responsive factor 2* (*Oserf2*) developed shorter roots than the wild type [211]. Abscisic acid has a significant role under moisture stress as it regulates the expression of several genes to restrict water loss and at the same time increase water uptake that enhances root growth. This root growth refers to the formation of lateral roots and root hairs, root tip swelling, and increasing water permeability [212]. This facilitates the uptake of water from deeper layers and maintains cell moisture level. A hypothetical illustration appears in Figure 3 to explain the interactive relationship between the nutrient transporters, hormonal relation with root types and the various genes controlling the development of different root types.

## 8. QTL Identification and Introgression of Root Architecture QTLs for DSR Using a Marker-Assisted Backcross Breeding Approach

Breeding for root architecture has been progressing slowly vis-à-vis breeding for aboveground plant traits in DSR due to the time-consuming and laborious phenotypic selection for root traits. Designing RSA suitable to adverse dry conditions can be a solution to the constraints in DSR. As the environment highly influences root phenotypes, selection programs must also be accompanied by stability analysis or genotype × environment (G×E) interaction studies [213]. The importance of genomic regions of QTLs and genes that are associated with RSA traits were depicted in Figure 4. One of the most common approaches has been to employ markers for the selection of the desirable genes and QTLs while screening as well as using introgression. Employing marker-assisted selection (MAS) procedures for the introgression of QTLs and genes can be an easier way, save time, and facilitate indirect selection, which can then be followed by phenotypic selection. MAS is already a widely used molecular technique to support rice breeding programs for the formation of divergent base populations, construction of genetic maps [214], identification of QTLs, identification of functional markers [215], as well as genomic selection [37].

The initial step is always to explore or screen useful genotypes and their concerned genic regions in a germplasm accession, landrace, or even a wild species for further transfer in a useful genetic background. After the identification of these potential genes/QTLs in germplasm or wild resources, if we are able to locate useful rare alleles from an elite or improved variety using markers (gene-specific/flanking), then the breeding program becomes much easier. For instance, after the reports published about the *Pup1* allele for P deficiency, a screening program revealed that out of the 96 genotypes (that included upland cultivars and landraces), 76 genotypes showed the presence of *PSTOL1*. N22, Dinoroda, Bowde, Bamawpyan, Tepiboro, Karni, Lalsankari, Surjamukhi, Hazaridhan, and KalingaIII marked positive for two of the closest flanking markers and two gene-specific markers for *Pup1* [216]. Additionally, the report of the Indian mega-variety Swarna possessing *OsPSTOL* (*Pup1*) eased the process of marker-assisted backcross breeding. ICAR- Indian Institute of Rice Research, Hyderabad used Swarna as a donor parent to introgress Improved Samba Mahsuri (loaded with bacterial blight resistance genes) with the *Pup1* candidate gene *OsPSTOL* [217]. To enhance nitrogen- as well as phosphorus-use efficiency, a backcross program is currently being carried out in NRRI, Cuttack. Researchers have used nitrogen-use-efficient CR Dhan 310 as the recurrent parent and low-P-tolerant CR Dhan 801 as the donor. This program has so far identified a genotype of the BC_1_F_6_ generation to be positive for *PSTOL1*, *DTY 1.1*, *DTY 3.1*, and *Sub1* in the background of nitrogen-use-efficient CR Dhan 310 [218]. A multi-stacking program was used to load the popular cultivar Lalat MAS (*xa5*, *xa13*, and *Xa21*) with eight different QTLs that included *DTY1.1*, *DTY 2.1*, *DTY 3.1*, *Sub1*, and *PSTOL1*. After genotyping the lines with SNP and following MAS, line number 48 of the BC_3_F_2_ generation had all the desired QTLs except *qDTY 3.1* [218]. In a program for screening the deep rooting QTLs *Dro1* and *Dro2* in 348 germplasm lines selected based on their root angle expression, 11 genotypes (Dular, Tepiboro, Surjamukhi, Bamawpyan, N22, Dinorado, Karni, Kusuma, Bowdel, Lalsankari, and Laxmikajal) have been reported to be positive for both QTLs [219]. Arai-Sanoh et al. [220] also observed higher grain yield with enhanced nitrogen uptake in *Dro1*-NIL lines than in IR64 owing to the deeper rooting. Successful examples of rice cultivars developed through MABC include “Birsa Vikas Dhan 111” with QTLs from the donor Azucena that expressed longer roots, significant improvement in grain yield suitable for upland aerobic conditions [122], and salinity tolerance, and Pusa Basmati 1121 pyramided with the *Saltol* QTL [221]. Selvi et al. [222] have pyramided lines with maximum root length QTLs (*qRT11-7* × *qRT18-1* + *7-32*) suitable for resource-limited environments. MAS has also been used for selecting novel or candidate alleles responsible for aerobic adaptation under reproductive-stage drought stress [223,224,225,226]. Three major consistent-effect QTLs (*qDTY_1.1_*, *qDTY_1.3_*, and *qDTY_8.1_*) for grain yield and reproductive-stage drought stress were reported by Catolos et al. [227] from the drought-tolerant donor Dular. Considering the drought stress conditions, QTLs associated with grain yield (*qDTY*2.2, *qDTY*3.1, and *qDTY*12.1) were introgressed into the line MR219, a popular Malaysian cultivar [228]. With an upland rice cultivar as a donor, Li et al. [227] used a doubled haploid population to identify QTLs for root number, length, thickness, and whole biomass after testing at three different locations. Kitomi et al. [115] identified putative QTLs for root length on chromosomes 2, 6, and 8. Of these, the significant ones were *QRO1* (a 1.7-Mb region on chromosome 2) and *QRO2* (an 844-kb region on chromosome 6). Both of these QTLs may be promising genetic resources in rice for improving root system architecture. A novel QTL was reported on chromosome 6, *qRL_6.1_*, which governed root length [229]. Two genomic loci (loci id1024972 and loci id4002562) from the mapping populations Aus276 and KaliAus have been identified as hotspots for root architectural plasticity, and locus id7001156 showed a correlation of root architectural plasticity, and grain yield with the same SNP marker [230]. These reports offer encouraging results, and they should be exploited in designing a multi-genic QTL research program to bring all the desirable alleles of the concerned traits for DSR under one genetic background.

## 9. Hybrid Development for the Direct-Seeded Rice System by Altering RSA

Breeding and deploying climate-smart hybrids resilient to abiotic stress can be an answer to the prevailing global climatic alterations [231] and also supplement food security. Hybrids usually have a higher seeding rate and nutrient and water demand, which deem them unfit for direct-seeded conditions. However, when a vigorous hybrid possessing an ideal RSA to enhance uptake of nutrients and moisture is adapted to dry DSR conditions, this would greatly improve its performance across environments. In a sense, when we supplement the aboveground vigor of a hybrid with underground root traits like long thick roots, larger root plate, and optimum surface, as well as deep roots along with the discussed genes and transporters, then we can extend the use of hybrids beyond their traditional areas of cultivation.

At present, the cultivars bred for TPR conditions are being used in DSR conditions, but they often fail to reach the potential yield of the dry direct-seeded system of cultivation [13]. This may be attributed to the adaptability of the TPR varieties to the high moisture content in the soil and readily available forms of nutrients (such as ammonium) in contrast to the DSR system, where moisture is the major lacuna. Thus, designing a hybrid development program to produce cultivars better suited to the DSR system should be the focus at this time. The first step would be developing parents with specific traits that suit DSR cultivation that can be employed in the hybrid production system. When we dissect the traits required for hybrid development, the most important is water requirement or uptake. Here, the solution is to exploit the optimum from the minimum resources available. This would require all the root traits adapted to water stress, including longer root length (*DRO1*), root angle spread at 45 degrees, well-branched fine roots, and an ample amount of root hairs. Deeper roots can help in moisture uptake and improve grain yield and NUE, and impart lodging tolerance under the DSR system of cultivation [163]. This review has already shed light on the genes regulating these traits, and their fixation in inbreds or parental lines should be a prerequisite for the DSR hybrid chain. The next equally important need of a hybrid crop is an adequate amount of nutrient supply. Anand et al. [232] evaluated hybrids for DSR in the Thungabhadra project command area of India. They suggested that, with a 25% higher rate of recommended fertilizer, hybrid KRH-4 performed well. As the DSR system hinders a continuous nutrient supply and is certainly not a high-input system of cultivation, we should look for the scope to exploit the available resources to the maximum. Traits that need attention at this point are well-spread surface roots (for immobile nutrients such as P, Fe, and Zn at the early growth stages), deeper roots at the later growth stage (for nitrogen uptake), higher carbon source accumulation in the roots (for facilitating grain filling), and new root development in the later part of growth to meet nutrient demand by the plant. Sowing rates also represent a conflict in the practice of using hybrid seeds in DSR. According to findings by Sun et al. [233], the grain yield performance of hybrids was not affected when the seed rate was changed from 240/m^2^ to 60/m^2^ in the central Chinese region. On the contrary, the same study reported a decrease in grain yield with a decrease in seed rate in the inbreds. The hybrids with higher tillering capacity and spikelets per panicle might have compensated for the low sowing rate. Additionally, findings supported the superiority of hybrids over inbreds at a low seeding rate (80/m^2^); it was also observed that N accumulation and dry matter accumulation were higher in the hybrids [234]. Thus, the study suggests that N uptake is comparatively higher in hybrids under low seeding rates, and they are efficient in allocating nutrients to the reproductive organs. Along with this vigor, both the seedling and post-vegetative stages are important for yielding a hybrid’s maximum potential. Different markers for these traits have already been reported; for instance, *OsPupK46-2*, a gene identified closely associated with the *Pup1* locus in Kasalath, can be used as a marker [235], and similarly, *DRO1* is flanked by markers RM24393 and RM7424 (reference cultivar Nipponbare) [101]. These markers can be employed in marker-assisted selection in the parents that are to be used in rice hybrid seed production for DSR. However, not much effort has been made in this direction. In an attempt to test hybrids under DSR conditions, the lowland hybrid Magat (IR64616H) was grown under DSR and it yielded 5.3 t/ha [236]. To increase yield, nutrient, moisture uptake, and stability should be tested and bred for aerobic conditions before hybrids are released for cultivation.

## 10. Transgenic Breeding for Root System Architecture

Transgenic breeding so far has been least explored for root traits and their improvement. This may be because of the lack of understanding of the genes involved and because of the pleiotropic effects of the concerned traits [237]. Most of the progress in transgenics has been made in the receptor kinase *PSTOL1*, overexpression of transcription factors *OsMYB2* and *OsNAC5*/*9*, root architecture associated gene (*OsRAA1*), expansin gene *OsEXPA8*, and deep root system gene *DRO1*. Studies have also been carried out on alteration of RSA, which increases N-, P-, and water-use efficiency (WUE), ultimately increasing grain yield [38]. The introgression of the deep rooting allele *DRO1*, following both traditional and transgenic approaches, confirmed the role of a steep deep root system in increasing yield under drought. Studies on *OsPT1* expression using a constitutive promoter led to an increase in P content in tissues as compared to the wild type, but these plants were shorter and had a higher tiller number [238].

Apart from the earlier discussed root-related QTLs, phytohormones also play a role in regulating root genetic architecture. Auxin regulates the crown roots and seminal root growth, and this information was further supplemented by studies using rice microRNA (miR393) that hampers normal seminal and crown root growth [237]. This was due to the negative regulation of the *Arabidopsis* auxin receptor homolog gene *TRANSPORT INHIBITOR RESPONSE 1* (*TIR1*) and *AUXIN SIGNALING F-BOX 2* (*AFB2*), *OsTIR1*, and *OsAFB2* [237]. Auxin activity is regulated with the help of *Aux*/*IAA* and *AUXIN RESPONSE FACTOR* (*ARF*) [239]. Mutant transgenics with a loss of function in the gene *OsIAA3* resulted in decreased crown root number. This gene was involved in the degradation-related domain, in which a conserved amino acid residue was targeted [240]. Similar to these reports, mutant studies on *crl1*/adventitious rootless1 (*arl1*) also revealed their role in crown root growth: the wild-type *CRL1*/*ARL1* gene encodes a *LATERAL ORGAN BOUNDARIES DOMAIN* (*LBD*)/*ASYMMETRIC LEAVES2-LIKE* (*ASL*) transcription factor acting downstream of the aforementioned *Aux*/*IAA-* and *ARF*-mediated pathway [88,241]. The same mechanism (involving *Aux*/*IAA* and *ARF*) is also seen in the case of *CRL6* encoding a chromodomain helicase DNA-binding (CHD) protein influencing crown root development, wherein most of the *Aux*/*IAA* genes are downregulated [42]. Even if the candidate gene for the mutant effect of *crl2* has not been identified, its relation to auxin signaling cannot be ruled out [203]. Besides auxin studies, cytokinin studies have been carried out, which shed light on the role of cytokinin in crown root development. A dominant mutant, root enhancer1 (*ren1-D*), produces a higher number of crown roots due to the expression of a *CYTOKININ OXIDASE*/*DEHYDROGENASE* (*CKX*) family gene, *OsCKX4* [209]. Interestingly, cytokinin also inhibits root length; in such cases, *CKX’s* enzymes can be helpful in the irreversible degradation process of cytokinin [242]. Similarly, the gene *METALLOTHIONEIN 2b* (*OsMT2b*) also regulated crown root and lateral root development by controlling rice plants’ cytokinin levels [243]. It is noteworthy to mention that cytokinin signaling affects crown root formation in plants. Further analysis has demonstrated that, along with auxin and cytokinin, *ethylene-responsive factor* (*ERF*) also regulates root structure [244]. Expression studies on *O. sativa ROOT ARCHITECTURE ASSOCIATED 1* (*OsRAA1*) showed enhanced crown root and lateral root numbers [245]. This gene is involved in cell cycle regulation; it is an anaphase-promoting cyclosome complex at the transition checkpoint between metaphase and anaphase [246]. *CULLIN-ASSOCIATED AND NEDDYLATIONDISSOCIATED 1* (*OsCAND1*), a cell division-related gene regulating the emergence of crown roots, has been found to be a *SCFTIR1 E3* ubiquitin ligase, carrying out the ubiquitination of auxin-associated proteins, thereby hampering root growth in *Arabidopsis thaliana* [247,248]. T-DNA insertion mutation experiments with the gene *AUXIN RESISTANT 1* (*OsAUX1*) exhibited a decrease in lateral root number and this was found to be related to the *AUX1*/*LIKE AUX 1* (*LAX*) gene family [244]. The altered expression of *OsPIN1* and *OsPIN2*, relating to endogenous levels of IAA, produced many fewer lateral roots, as seen in the double mutants *nal2* and *nal3* [90]. The underlying genes were found to be *OsWOX3A*/*OsNARROW SHEATH* (*OsNS*) [90]. Transgenic approaches combined with biotechnological tools have led to an increased understanding of the underlying QTLs and mechanisms. There is a further need to use such technology to develop superior cultivars with ideal RSA genes.

## 11. Modeling DSR with Root-Specific Traits

The development of varieties for DSR conditions can be accelerated by selecting suitable traits associated with root traits, such as seedling emergence, early vegetative vigor, nutrient uptake, nodal root number, root hair length, and density under rainfed conditions [70]. Even though several QTLs have been identified for RSA in rice, an ideal root model for DSR is still lacking. No complete information exists about the ideal root spreading angle and type of roots needed for different nutrient uptake, whether deep rooting alone is enough to combat drought stress, early root vigor, and root structure for lodging tolerance. A suggestive model for ideal RSA has been proposed by Kitomi et al., [115], but no such concepts have been put forward in relation to rice RSA in the dry or aerobic system of rice cultivation. Therefore, we amalgamate all the available information from diverse research works on root traits to develop and propose a root model for DSR. The desirable traits and the concerned genes/transporters expressed in the roots and that have shown promising effects (cited in previous sections) are included in our model. The foremost trait is early root vigor, referring to a root system with all the necessary root traits required by the plant at an early stage to provide the shoot with the optimum level of nutrients and moisture along with a uniform stand. The gene *PSTOL1*, an enhancer of early root growth, can facilitate this. QTLs such as *qEUE11.1* and *qEVV9.1* can also aid in the course. The next is the seedling crown root and nodal roots: a higher number of crown roots, as well as nodal roots, are required to uptake immobile soil nutrients such as P, Fe, and Zn from topsoil. The QTL *qSOR1* would also produce more surface rooting. On the nutrient uptake front, high-affinity P transporters are required as early as 2 DAG, and *OsPT1* and *OsPT8* genes can aid in P uptake from 2 to 3 DAG. Unlike for other nutrients, N requirement and uptake continue throughout the crop growth stages, so nitrate transporters active during the entire life cycle are more desirable. Nitrogen transporters such as *OsNPF7.2* and *OsAMT1;3* can be used for nitrate and ammonium, respectively. Nitrogen uptake in the later stages of plant growth would require deep roots, and candidate gene *DRO1* would satisfy the need for both deep roots and exploiting root angles to generate surface roots. A Fe transporter/Fe deficiency-tolerant gene needs to be identified and introgressed. Low xylem diameter for lower hydraulic conductivity with more conservative water-use efficiency and the maximum number of roots should be within a 45^o^ spreading angle. Inverted V-shaped roots are suitable for a DSR deep-dimorphic root system. This new root development at the post-anthesis stage can also aid in proper supply to cater to the high nutrient demand by the growing reproductive organs. During this stage, grain filling would rely on carbon sources present in the vegetative parts of the plant, and thus a higher amount of carbon in the roots is desirable. High root density with fine root hairs can aid in the uptake of moisture from the finest of the pores. The genes and QTLs involved in these post-heading vigor traits are yet to be identified. Higher root number and density would also enhance the root plate, thus imparting lodging tolerance. The QTLs *qLDG_3.1_* and *qLDG_4.1_* identified in relation to this trait should also be exploited. Already, QTL information has been confirmed for most of the abovementioned traits. Therefore, the further need is pyramiding desirable QTLs together into a required genotype to develop a range of superior cultivars with suitable root architecture for DSR to meet future rice demand under the scenario of climate change.

## 12. Conclusions and Future Perspectives

To meet food security and future rice demand, especially under changing climatic conditions, there is an urgent need for an environmentally sustainable strategy. For this, DSR is the best alternative instead of conventional puddled transplanted rice because the former is advantageous in water-saving and labor-saving. Among the various agro-morphological traits, RSA traits are the major component traits in the DSR ecosystem to boost productivity. To achieve this target, it is crucial to understand the genetic and molecular mechanisms that govern root system architecture in rice. Using this information for identifying RSA traits that best complement the aboveground plant architecture can enhance productivity under DSR conditions. In addition, understanding the hormonal cross-talk (mainly auxin/cytokinin) regulations and signaling pathway coordination between the root and shoot system is essential for increasing efficiency in transport mechanisms. Root geometry and architectural changes in response to environmental challenges determine overall productivity, performance, and fitness. The advances in molecular breeding technologies such as marker-assisted selection, genome sequencing, CRISPR/Cas-9-mediated genome editing, and TILLING approaches provide an opportunity to dissect and identify the novel QTLs and genes/alleles related to RSA traits in the DSR ecosystem. The MAS strategy is the best and most cost-effective technique over conventional breeding approaches by enhancing the precision and efficiency of DSR improvement through the introgression of root-associated QTLs. Innovative high-throughput root phenotyping platforms provide a new step toward filling the gap between field and laboratory analysis of root system architecture to correlate whole-plant growth with yield. These techniques will help achieve a deeper understanding of specific root traits, their recipient and donor parents, and genetic markers for improving yield through the development of root traits. The imperative root traits such as the number of crown roots and adventitious roots, root length, and root spreading angle are fundamental for incorporating into future breeding lines to enhance water and nutrient acquisition and at the same time, maintain yield. Therefore, this requires careful introgression of significant QTLs into desirable genotypes to increase yield with efficient water and nutrient use under stressful environments. These root-related traits with high-throughput techniques will support developing high-yielding, resource-efficient DSR varieties with wider adaptability.

## Figures and Tables

**Figure 1 ijms-22-06058-f001:**
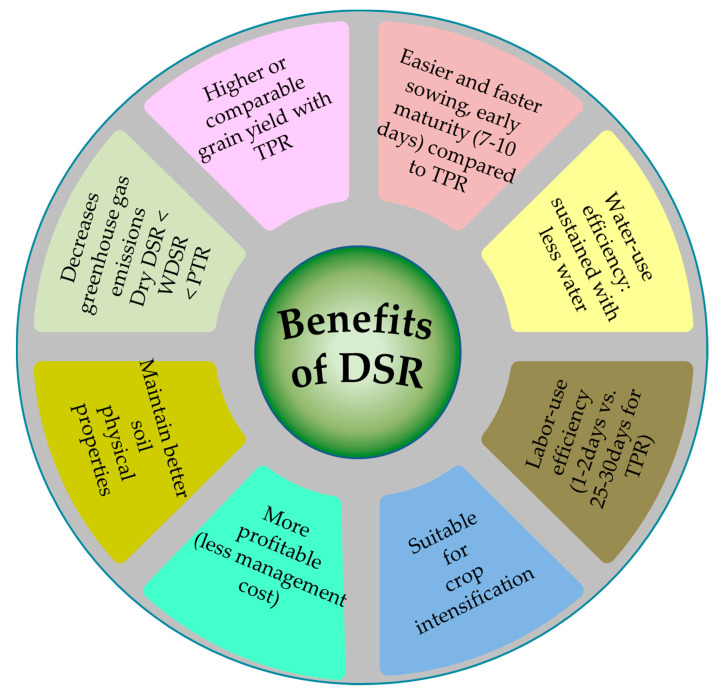
Major benefits of direct-seeded rice (DSR) system over transplanted rice (TPR) system.

**Figure 2 ijms-22-06058-f002:**
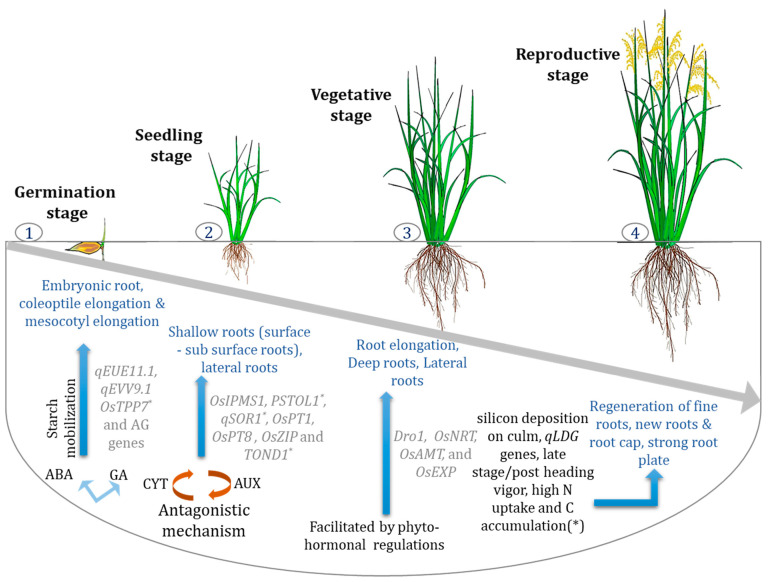
Ideal DSR root system with root-specific traits and genes/QTLs at different stages of growth. The varying growth stages have different requirements of root morphology and associated genes/QTLs depicted in the Figure. (**1**) The germinating seed requires AG genes for germination: *qEUE11.1* and *qEVV9.1* for early high seedling vigor and *qSOR1* for better surface rooting. (**2**) From the nutrient perspective, the seedling stage would require *OsPT1* and *OsPT8* for P uptake, *OsZIP* genes for Zn uptake, and *TOND1* for nitrogen deficiency tolerance, supplemented with *PSTOL1* (a root growth enhancer). (**3**) The vegetative stage needs DRO genes that articulate the roots working in complementation with *SOR, OsNRT*, and *OsAMT* genes for nitrogen transport efficiency, expansin genes (such as *OsEXP*) for growth of root hair and increased root length, and most of the roots are ideally at a 45° angle with each other measured from the base. (**4**) The reproductive stage needs better anchorage and root spread, higher silicon deposition on culm supplemented with *qLDG* genes for lodging tolerance, higher nutrient and moisture uptake compensated with more fine roots, new roots and root cap development, high N uptake, and C accumulation. * The genes/QTLs for these attributes are yet to be identified.

**Figure 3 ijms-22-06058-f003:**
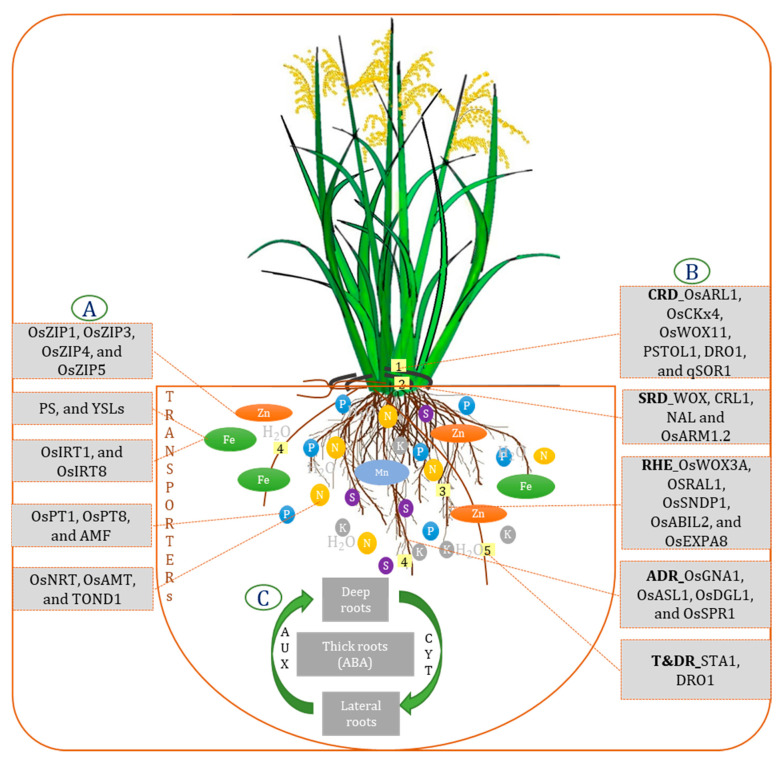
A schematic representation of the root growth zone and types of roots illustrated with necessary genes/QTLs for the DSR root system and their interaction with soil nutrients. The right side (**A**) indicates the essential nutrients and their transporters, and left side (**B**) indicates the specific root attributed traits along with their regulating genes/QTLs under the DSR system of rice cultivation, and the main center (**C**) indicates the hormonal regulation in development of different types of roots (CRD: Crown root development; SRD: Surface root development; RHE: Root hair elongation; ADR: Adventitious roots; and T&DR: Thick and deeper roots. Immobile nutrients (surface layer of the soil) Zn, Mn, Fe, and P are acquired by the crown roots, surface roots, and lateral roots. *PSTOL1*, *qSOR1*, and *DRO1* help in surface root development, whereas *WOX*, *CRL1*, and *NAL* genes and *OsARM1,2* stimulate crown root and lateral root growth. These roots proliferate when auxin levels are low in the plant, unlike deep roots that require a higher auxin level. Thick roots are developed with the help of gene *STA1* and when the abscisic acid level is high. Besides Zn and Fe transporters, phytosiderophores (root exudates) and *YSLs* play a role in Zn and Fe uptake. AMF colonizes in the roots, enhancing P uptake by the plant along with the respective transporters. [PS: phytosiderophores; AMF: arbuscular mycorrhizal fungi; *OsPT1* and *OsPT8*: phosphorus transporters; *OsZIP1*, *OsZIP3*, *OsZIP4*, and *OsZIP5*: zinc transporters; *OsIRT1* and *OsIRT2*: iron transporters; *OsAMT*: nitrogen (ammonium) transporter; *OsNRT*: nitrogen (nitrate) transporter; YSL: yellow stripe-like protein; TOND1: tolerance of nitrogen deficiency 1; *PSTOL1*: phosphorus starvation tolerance 1; *qSOR1*: soil surface rooting 1; *DRO1*: deeper rooting 1; *WOX*: Wuschel-related homeobox gene; *CRL1*: crown rootless 1; NAL: narrow-leaf; *OsARM1,2*: armadillo proteins; STA1: stele transversal area 1; AUX: auxin; CTY: cytokinin; ABA: abscisic acid]. Arrows indicate the increasing/decreasing concentration of growth hormones and their involvement in root development.

**Figure 4 ijms-22-06058-f004:**
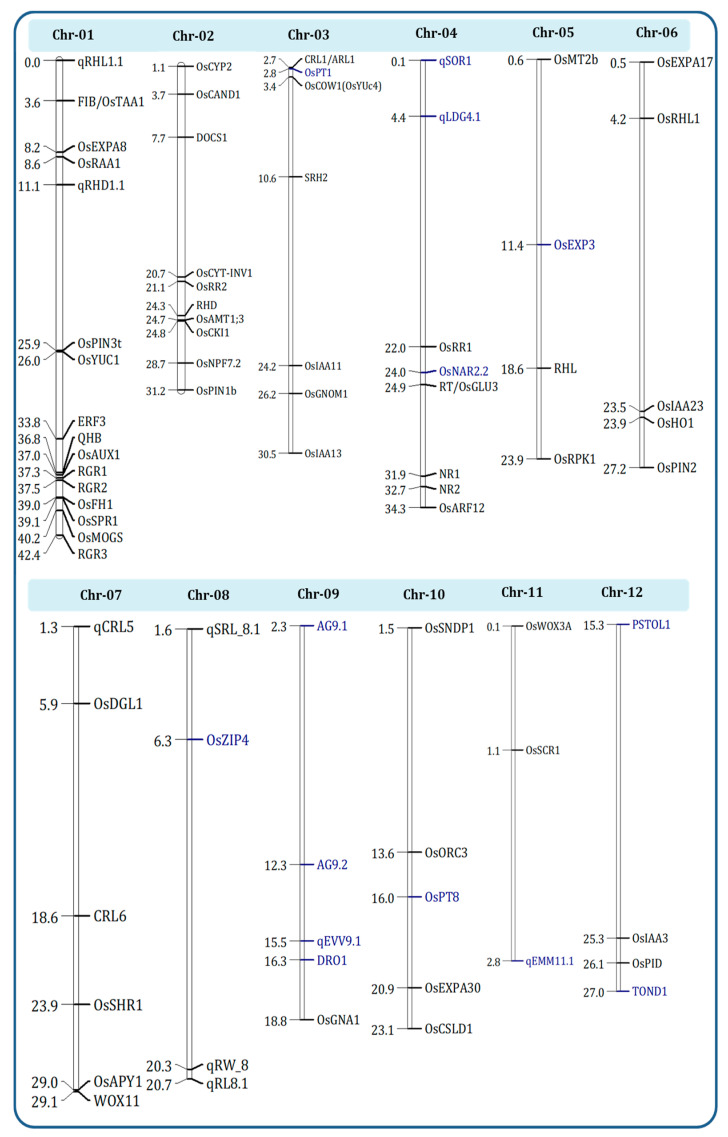
Depicted diagram of the genomic regions of root system architecture (RSA) traits associated with QTLs and genes in all chromosomes. The left bar numeric values indicate the genomic position (in Mb), and the names of QTLs and genes are on the right side. The blue color represents the important QTLs and genes used in improving RSA traits in DSR conditions, whereas the black color represents the QTLs and genes collected from the comprehensive literature survey on RSA traits under DSR conditions.

## Data Availability

Not applicable.

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
