# Peer review of "Proofing Direct-Seeded Rice with Better Root Plasticity and Architecture"

_ijms, 2021, doi:10.3390/ijms22116058_

Round 1

Reviewer 1 Report

In this manuscript author discussed about the proofing direct-seeded rice with better root plasticity and architecture. In this review, authors tried to mine the available research information on the direct-seeded rice (DSR) root system to highlight the requirements of different root traits such as root architecture, length, number, density, thickness, diameter, and angle that play a pivotal role in determining the uptake of nutrients and moisture at different stages of plant growth. RSA also faces several stresses due to excess or deficiency of moisture and nutrients, low or high temperature, or saline conditions. To counteract these hindrances, adaptation in response to the stress becomes essential. Candidate genes such as early root growth enhancer PSTOL1, surface rooting QTL qSOR1, deep rooting gene DRO1, and numerous transporters for their respective nutrients and stress-responsive factors have been identified and validated under different circumstances. Identifying the desired QTLs and transporters underlying these traits and then designing an ideal root architecture can help develop a suitable DSR cultivar and aid in further advancement in this direction.

I want to congratulate the authors for writing on a vital topic. The manuscript will attract a large readership. The manuscript is written very well, and I found no major drawback in it. But for the betterment of this review, I have some feedback. Please refine the manuscript accordingly.

Change at

L151 with higher yield to with a higher yield.

L256 What are the full form of IAA, ABA, and JA.

L347 which regulate LR development to which regulates LR development.

L406 What is (change correct?)?

L494 What is (of what? nutrient supply?)?

L499 need to be to needs to be.

L647 om chromosome 8 to on chromosome 8.

L658 root trait expression either directly to root trait expressions either directly.

L663 radicleless or radicle less?

L739 revealed that, out of the 96 to revealed that out of the 96.

L794 as thick long roots to as long thick roots.

L806 and ample amount of root hairs to and an ample amount of root hairs.

L913 as well as nodal roots is required to as well as nodal roots are required .

L942 in water saving and labor saving to in water-saving and labor-saving.

I have found some plagiarism at L120-L221, L204-205, L244-245, L249-250, L264-265, L440-443, L477-478, L605-606, L617-618, L669-670. Please clean it.

Author Response

Dear Managing Editor and Editorial Board Members

International Journal of Molecular Science (IJMS)

Thank you so much for providing your feedback and suggestions for our manuscript “Proofing direct-seeded rice with better root plasticity and architecture (Manuscript ID:ijms-1216197))”. We are very grateful to the two anonymous reviewers for their time in the critical review and thoughtful comments on this manuscript. We have now implemented their suggestions and comments in the revised manuscript using the Track changes functions and highlighted red font. Our responses to reviewers' comments were provided below point-by-point with a highlighted blue color explaining how we are have addressed each section, according to the reviewer comments. We hope that the revised manuscript is now acceptable for publication in your esteemed journal as soon as possible.

Reviewer 1

The comments are below

Response to Reviewer 1: Thank you so much for your insightful comments and valuable suggestions to improve our manuscript. We have now incorporated the required information, and those changes are highlighted with a red color font in the revised manuscript. Please see below, in blue, for a point-by-point response to Reviewer 1 comments. We strongly appreciate the positive comment and suggestions.

Query 1: L151 with higher yield to with a higher yield.

Response: Complied as suggested by the reviewer, changed “with higher yield” to “with a higher yield” (Line no. 150)

Query 2: L256 What are the full form of IAA, ABA, and JA.

Response: Amended as suggested by the reviewer as “Indole acetic acid, Abscisic acid, and Jasmonic Acid” (Line no 253)

Query 3: L347 which regulate LR development to which regulates LR development.

Response: Amended as suggested by the reviewer, changed “which regulate LR development” to “which regulates LR development” (Line no 353)

Query 4: L406 What is (change correct?)?

Response: Amended as suggested by the reviewer. Error happened while editing for submission. (Line no 409)

Query 5: L494 What is (of what? nutrient supply?)?

Response: Amended as suggested by the reviewer. Error happened while editing for submission. (Line no 499)

Query 6: L499 need to be to needs to be.

Response: Amended as suggested by the reviewer, changed “need to be” to “needs to be” (Line no 504)

Query 7: L647 om chromosome 8 to on chromosome 8.

Response: Amended as suggested by the reviewer. “om chromosome 8” corrected as “on chromosome 8” (Line no 657)

Query 8: L658 root trait expression either directly to root trait expressions either directly.

Response: Amended as suggested by the reviewer. “root trait expression either directly” corrected as “root trait expressions either directly” (Line no 668)

Query 9: L663 radicleless or radicle less?

Response: radicleless (ral1) is the correct form of the gene name (Line no 673)

Query 10: L739 revealed that, out of the 96 to revealed that out of the 96.

Response: Amended as suggested by the reviewer. “revealed that, out of the 96” changed as “revealed that out of the 96” (Line no 749)

Query 11: L794 as thick long roots to as long thick roots.

Response: Amended as suggested by the reviewer. “as thick long roots” corrected as “as long thick roots” (Line no 802)

Query 12: L806 and ample amount of root hairs to and an ample amount of root hairs.

Response: Amended as suggested by the reviewer. “and ample amount of root hairs” corrected as “and an ample amount of root hairs” (Line no 817)

Query 13: L913 as well as nodal roots is required to as well as nodal roots are required.

Response: Amended as suggested by the reviewer. “as well as nodal roots are required” corrected as “as well as nodal roots are required” (Line no 926)

Query 14: L942 in water saving and labor saving to in water-saving and labor-saving.

Response: Amended as suggested by the reviewer. “in water-saving and labor-saving” corrected as “in water-saving and labor-saving” (Line no 956)

Query 15: I have found some plagiarism at L120-L221, L204-205, L244-245, L249-250, L264-265, L440-443, L477-478, L605-606, L617-618, L669-670. Please clean it.

Response: Complied as per the reviewer’s suggestions in respect of each section and edited accordingly.

Reviewer 2 Report

Dear Authors,

Your review is interesting and certainly regrouping a lot of knowledge around root architecture and other trait of interest. Therefore it would be useful addition to field. 

However, in my opinion, certain points would need revisiting to improve the quality of your paper. 

Here are my comment according to line numbering:

110: In your introduction you don't bring up your predictive model that is one of your final point. I would try to bring it up, but i have other comment on this point later.

170: Figure2. This figure is of quite poor quality and could probably be made clearer and more explicit. i tine line could be added and more detail on the kind of root could help too.

196: the way this sentence is written is confusing, i would advice to break it down and make it clearer. maybe bring some clarification about the depth of sowing you refer too, because between broadcasting and drilling, the critical traits for seedling vigour might be different.  

311: I think here you can't avoid to treat briefly the work done on mesocotyl elongation. Faster elongation can bring a major advantage to ride seedling compare to weeds. a brief resume of this field of research would be appreciated.

371:your anaerobic germination point is interesting, i would make a point of saying that well known cultivar (and still popular with farmers around the world) IR64, is sensitive to anaerobic condition and is inhibited.

maybe refering this paper https://www.ncbi.nlm.nih.gov/pmc/articles/PMC6711729/ 

406:  you should avoid this kind of remaining comment

494: you should avoid this kind of remaining comment

500-501: please elaborate this argument, this is quite strong statement.

507: a screen of a subset of IRRI collection could also bring some insight.

603: again the work around tolerance to deep sowing and the benefit this could have on lodging and water access should be brought up.

625: The presence of aerenchyma in rice root could be discuss briefly here in the context of drought etc.

695: very confusing and poor quality schematic. this is really not helping the reader to get a grip of the whole problematic treated here. Maybe split the schematic could help or re-draw.

744: should it be ICAR-IRRI or ICAR-NRRI?

799: maybe bring the man conclusion on why the cultivars fail in DSR rather than letting the reading seeking for it in the references.

853: poor English

906: your root model for DSR would be quite interesting for the community, is there a link to a publication describing it? It is difficult to take this statement seriously without proof that your model work as per gold standard. I would invite you to moderate this statement or to indicate the link to the model. 

934: In the context of climate change, i can't see the advantage to have ONE superior cultivar. In my point of view it would be better to create a range of cultivar capable of high yielding in different environment. Climate change might be global but the effect on lands and cultivation conditions are numerous, hence ONE superior cultivar would probably fail to deliver in all scenarios. This is a bit what you end by in line 969, so maybe you avoid to make a counter argument before...

some DOI (if available) are missing in references section for the ref

8/ 58/113/117/199/214/221/222/226

Author Response

Reviewer 2

Response: Thank you very much for your valuable comments and suggestions. According to your suggestions, we have provided detailed information and divided the point-wise to respond in the best possible way.

Query 1: 110: In your introduction you don't bring up your predictive model that is one of your final point. I would try to bring it up, but i have other comment on this point later.

Response:  Complied as per the reviewer’s suggestion. In addition, we have suggested a model for an ideal RSA with necessary genes/QTLs suitable for DSR system.”  (Newline L113-115).-

Query 2: 170: Figure2. This figure is of quite poor quality and could probably be made clearer and more explicit. i tine line could be added and more detail on the kind of root could help too.

Response: We have now improved Figure 2 for a better and clear version as per the comments of the reviewer

Query 3: 196: the way this sentence is written is confusing, i would advice to break it down and make it clearer. maybe bring some clarification about the depth of sowing you refer too, because between broadcasting and drilling, the critical traits for seedling vigour might be different. 

Response: Irrespective of sowing methods (broadcasting or drilling), the initial period of seedling growth requires a surface rooting trait for uptake of nutrient like P for root growth, which initiated P on the 3rd after sowing. In the case of deep sowing, higher expression of mesocotyl and coleoptile elongation genes are required. Besides, the deep setting of seeds also helps in increasing lodging tolerance and deeper roots access to soil moisture from zones that otherwise would be inaccessible by the plant. We have amended the sentence as “The key player in developing different type of root system in the early seedling growth are the genes controlling root angle, which determines the development of surface and shallow subsurface roots.” (Line no. 195-197)

Query 4: 311: I think here you can't avoid to treat briefly the work done on mesocotyl elongation. Faster elongation can bring a major advantage to ride seedling compare to weeds. a brief resume of this field of research would be appreciated.

Response: Mesocotyl, the portion between the coleoptile node and the base of the seminal root, is one of the key players in pushing the germinating seedlings out of the soil. Its length determines the seedling emergence and establishment. GWAS study for mescotyl elongation in 208 accessions of rice has identified six novel loci explaining up to 15.9% phenotypic variations (Hongyan Liu et al. 2020). Three QTLs, namely qMel-1, qMel-3, and qMel-6 controlling mesocotyl length, have been reported on chromosomes 1,3, and 6 respectively in backcross inbred lines developed from a cross between Kasalath and Nipponbare (Lee et al. 2017). (Line No. 310-316)

Query 5: 371: your anaerobic germination point is interesting, i would make a point of saying that well known cultivar (and still popular with farmers around the world) IR64, is sensitive to anaerobic condition and is inhibited. maybe refering this paper https://www.ncbi.nlm.nih.gov/pmc/articles/PMC6711729/

Response: The reviewers mentioned as anaerobic germination (AG) tolerance is an important trait for the DSR condition. Long duration (145-160 days), rainfed lowland rice cultivars can germinate without oxygen as they can synthesize the enzymes required for starch degradation even in oxygen-depleted soil layers. However, a popular rice variety like IR64 (110 days duration) developed for transplanted irrigated ecosystems does not possess AG. Due to the switching of rice cultivation from TPR to DSR condition, irrespective of duration, all genotypes should possess AG traits to overcome water stagnation due to poor leveling and heavy rainfalls immediately after sowing.

Query 6: 406:  you should avoid this kind of remaining comment

Response: Amended as suggested by the reviewer. Error happened while editing for submission. (Line no 409)

Query 7: 494: you should avoid this kind of remaining comment

Response: Amended as suggested by the reviewer. Error happened while editing for submission. (Line no 499)

Query 8: 500-501: please elaborate this argument, this is quite strong statement.

Response: Complied as per the reviewer’s suggestion.

The problem of salinity or salt stress has been increasing gradually, resulting mostly from faulty agricultural practices (frequent light irrigation, higher use of fertilizers such as muriate of potash and ammonium sulphate, summer fallowing etc.) (Line no. 550 – 551)

Query 9: 507: a screen of a subset of IRRI collection could also bring some insight.

Response: We have already cited the insights of the salinity stress and physiological elements in the revised manuscript.

Query 10: 603: again the work around tolerance to deep sowing and the benefit this could have on lodging and water access should be brought up.

Response: Complied as per the reviewer’s suggestion.

It is noteworthy that deep sowing in direct seeded rice has also become a phenomenon wherein it helps the plant have better anchorage and imparts lodging tolerance. This would require a higher expression of mesocotyl and coleoptile elongation genes (Lee et al., 2017). At the same time, deep setting of seeds also helps in deeper roots and access to soil moisture from zones that otherwise would be inaccessible by the plant. (Line no. 506 – 507)

Query 11: 625: The presence of aerenchyma in rice root could be discuss briefly here in the context of drought etc.

Response: Complied as per the reviewer’s suggestion.

Apart from the above, the recent reports on the role of aerenchyma tissue that aids tolerance to moisture stress. The formation of lysogenous cortical aerenchyma cells helps the plant to reduce the metabolic cost and allowing for greater water uptake from soil (Zhu, Brown, and Lynch 2010). (Line no 487 -489)

Query 12: 695: very confusing and poor quality schematic. this is really not helping the reader to get a grip of the whole problematic treated here. Maybe split the schematic could help or re-draw.

Response : Figure 3 has been updated with a better and clearer version as per the reviewer’s comment.

Query 13: 744: should it be ICAR-IRRI or ICAR-NRRI?

Query 14: 799: maybe bring the man conclusion on why the cultivars fail in DSR rather than letting the reading seeking for it in the references.

Response : Amended as suggested by the reviewer. As “This may be attributed to the adaptability of the TPR varieties to the high moisture content in the soil and readily available forms of nutrients (such as ammonium) in contrast to the DSR system where moisture is the major lacuna”. (Line no 808-810)

Query 15: 853: poor English

Response: Amended as suggested by the reviewer.

Apart from the earlier discussed root-related QTLs, phytohormones also play a role in regulating root genetic architecture. (Line no 865)

Query 16: 906: your root model for DSR would be quite interesting for the community, is there a link to a publication describing it? It is difficult to take this statement seriously without proof that your model work as per gold standard. I would invite you to moderate this statement or to indicate the link to the model.

Response : Complied as per the reviewer’s suggestion.

A suggestive model for ideal RSA has been proposed by Kitomi et al. (2018), but no such concepts have been put forward in relation to rice RSA in the dry or aerobic rice cultivation system. (Line no. 916-918)

Query 17: 934: In the context of climate change, i can't see the advantage to have ONE superior cultivar. In my point of view it would be better to create a range of cultivar capable of high yielding in different environment. Climate change might be global but the effect on lands and cultivation conditions are numerous, hence ONE superior cultivar would probably fail to deliver in all scenarios. This is a bit what you end by in line 969, so maybe you avoid to make a counter argument before...

Response: Amended as suggested by the reviewer.

“QTL information has been confirmed for most of the traits as mentioned above. Therefore, the further need is pyramiding of desirable QTLs together into a required genotype to develop a range of superior cultivars with suitable root architecture for DSR to meet future rice demand under the scenario of climate change”. (Line no 947-951, 980, 982)

Query 18: some DOI (if available) are missing in references section for the ref

Response : Citations and bibliography have been set using the reference manager tool of Mendeley. However, the missing DOIs have been updated manually. The doi of few papers were not available, which is the reason for its absence in the bibliography. However, the other details of the publication, such as year, volume etc., updated.

Round 2

Reviewer 1 Report

I am happy with the response of the authors and think that this manuscript can be published in its current format